🔓 | **Open Peer Review** | Microbial Pathogenesis | Research Article

# Gut microbiota dysbiosis and amino acid metabolic dysregulation in children with allergic rhinitis complicated by constipation

Shen Lv,[1] Weikeng Yang,[2] Chunyun Sun,[3] Ying Huang,[4] Wei Kong,[1] Zhi Wan,[1] Congfu Huang[1]

**ABSTRACT**  This study investigates the pathogenesis of constipation in children with allergic rhinitis (AR) by analyzing gut microbiota (GM) composition and fecal metabolites. We enrolled 21 AR children (AR group) and 16 AR children with constipation (ARFC group), performing absolute quantitative 16S rRNA sequencing and non-targeted metabolomics on fecal samples. The key findings are as follows: (i) The ARFC group exhibited significantly elevated *Hungatella* abundance ($P = 0.033$) and reduced beneficial genera (*Bifidobacterium*, *Blautia*, etc.; $P < 0.05$). (ii) Fecal metabolomics revealed upregulation of indoxyl sulfate and downregulation of amino acid metabolites (e.g., aromatic and branched chain amino acids) in ARFC children, with enriched pathways in amino acid metabolism and nutrient absorption. (iii) Correlation analysis demonstrated a positive association between *Hungatella* and indoxyl sulfate, as well as between diminished probiotics and disrupted amino acid metabolism. These results suggest that *Hungatella* and indoxyl sulfate may serve as microbial and metabolic biomarkers for constipation in AR children. GM dysbiosis linked to amino acid dysregulation could drive constipation pathogenesis, highlighting potential therapeutic targets for probiotic or dietary interventions.

**IMPORTANCE** Allergic rhinitis (AR) is a common allergic disorder in children, often accompanied by gastrointestinal symptoms such as constipation, yet the underlying mechanisms remain poorly understood. This study reveals that constipation in AR children is associated with specific gut microbiota alterations—particularly an increase in *Hungatella* and a decrease in beneficial genera like *Bifidobacterium*—and disrupted amino acid metabolism, notably elevated indoxyl sulfate. These findings provide novel microbial and metabolic biomarkers for constipation comorbidity in AR patients. By highlighting the role of gut microbiota dysbiosis and metabolic dysregulation in driving gastrointestinal dysfunction, this work underscores the potential for microbiota-targeted interventions, such as probiotics or dietary adjustments, to improve both gut health and allergic outcomes in children.

**CLINICAL TRIALS** This study is registered with Chinese Clinical Trial Registry as ChiCTR2400085982.

**KEYWORDS**  allergic rhinitis, constipation, gut microbiota, metabolomics, *Hungatella*, indoxyl sulfate

**Peer Reviewer** Rachael Terumbur Duche, Joseph Sarwuan Tarka University Makurdi, Makurdi, Benue, Nigeria

Address correspondence to Congfu Huang, 78333755@qq.com.

Shen Lv, Weikeng Yang, and Chunyun Sun contributed equally to this article. The order of the co-first authors' names was determined by mutual agreement among the authors.

The authors declare no conflict of interest.

Allergic rhinitis (AR) is a chronic non-infectious inflammation of the nasal mucosa. It is primarily mediated by allergen-specific immunoglobulin E (IgE) following exposure to allergens in sensitized individuals. Additionally, non-IgE-mediated mechanisms and neuroimmune dysregulation are also implicated in its pathogenesis. Genetic

susceptibility is a key factor in AR, but the living environment and gut microbiota (GM) also play important roles in its development (1). In clinical research, we have observed that a significant proportion of children with AR experience gastrointestinal symptoms. These include constipation, anorexia, poor appetite, and diarrhea, among other gastrointestinal issues (2). Similar associations have been noted in adults with AR. For instance, female AR patients have more frequent medical visits for conditions such as gastritis/duodenitis, gastrointestinal functional disorders, gastroenteritis/colitis, and constipation (3). Constipation, in particular, can lead to several complications. These may include anal fissures, rectal bleeding, fecal retention, painful defecation, and emotional tension during bowel movements. Moreover, constipation can contribute to sleep disturbances. In school-age children, this can manifest as inattention, irritability, and hyperactivity. These symptoms can increase the frequency of outpatient visits or hospitalizations, thereby raising healthcare costs and significantly impacting the quality of life of affected children (4).

It is well established that GM dysbiosis is present in both individuals with AR (5–7) and those with constipation (8–10). In fact, GM dysbiosis may be more pronounced in AR children who also suffer from constipation. Studies have shown that in AR children with functional gastrointestinal diseases, the levels of beneficial bacteria such as *Bifidobacterium* and *Lactobacillus* are reduced, while the levels of potentially pathogenic bacteria like *Enterobacter* and *Enterococcus* are increased. Additionally, the levels of gastrointestinal peptides, including gastrin, motilin, somatostatin, and serotonin, are also altered (6). Interventions with glutamine and probiotics have been shown to optimize the GM composition (6). Functional gastrointestinal diseases encompass a range of symptoms, including constipation, diarrhea, and abdominal pain. These different symptoms can have varying impacts on the composition and function of GM. Therefore, further research is needed to explore the correlation between AR and different manifestations of functional gastrointestinal diseases in the context of GM. This will help guide the rational selection of probiotics for therapeutic purposes. Constipation may be linked to dysregulation of the mucosal immune system and dysfunction of the central nervous system. Abnormal secretion of gastrointestinal peptides and GM imbalance also play important roles in weakening gastrointestinal motility (11). Khalif et al. (12) observed that constipation is associated with significant changes in GM, intestinal permeability (often referred to as "leaky gut"), and immune response. They also noted that alleviating constipation often reverses these changes. Moreover, constipation may affect immunity by altering GM and increasing intestinal permeability. This can lead to the hypersecretion of proinflammatory or inflammatory biomarkers, such as chemokines and cytokines (13, 14). These changes can induce or exacerbate allergic symptoms, highlighting the complex interplay between gastrointestinal health and systemic immunity.

The current drug treatments for constipation face challenges such as poor efficacy and a high rate of recurrence. Therefore, it is crucial to investigate the pathogenesis of constipation and its related diseases, as well as to explore more effective treatment methods. These methods may include dietary interventions, GM regulation, and immunotherapy. Such approaches could be particularly beneficial for treating constipation in children with AR and for improving their quality of life. Currently, there is limited research on the gut microbiota and metabolic characteristics of children with AR who also have constipation. These comorbidities may exacerbate the condition through specific microbiota-metabolic axes, which urgently require further exploration. In this study, we used the absolute count of GM and non-targeted metabolomics technology. We compared and analyzed the differences in GM diversity and metabolite levels across different groups of children. We also examined the correlations between significantly different bacteria and metabolites, as well as the relationships between GM, metabolites, and clinical phenotypes. Based on these findings, we explored the effects and mechanisms by which constipation influences AR.

## MATERIALS AND METHODS

### Study design and participant recruitment

This cross-sectional study was conducted at the Pediatric Department of Longgang District Maternity and Child Healthcare Hospital (Shenzhen, China).

A total of 37 preschool-aged children were recruited between January and December 2024. Participants were divided into two groups: (i) AR group ($n$ = 21): Children diagnosed with AR per ARIA guidelines (1); (ii) AR children with constipation (ARFC) group ($n$ = 16): Children diagnosed with both AR and functional constipation according to Rome IV criteria (15). Exclusion criteria: History of severe gastrointestinal/hepatic/renal diseases, antibiotic/probiotic use within the past month, developmental disorders, or infectious diseases. Dietary habits, antibiotic exposure, and family environment were assessed via standardized questionnaires, with no significant intergroup differences ($P$ > 0.05). Demographic characteristics were compared using the Wilcoxon rank-sum and Fisher's exact tests.

### Fecal sample collection and storage

Fresh fecal samples were collected using sterile swabs (Copan Italia, Brescia, Italy), immediately transferred to cryotubes (Sarstedt, Nümbrecht, Germany), and stored at −80℃ within 30 min to preserve microbial and metabolic integrity.

### Absolute quantitative 16S rRNA gene sequencing

#### DNA extraction and quality control

Genomic DNA was extracted from 200 mg fecal samples using the FastPure Stool DNA Isolation Kit (Vazyme Biotech, Nanjing, China). DNA purity (A260/A280 ratio: 1.8-2.0) and concentration were measured using a NanoDrop2000 spectrophotometer (Thermo Fisher Scientific, USA). DNA integrity was verified via 1% agarose gel electrophoresis (100 V, 30 min).

#### Spike-in DNA addition and PCR amplification

Synthetic spike-in DNA (12 unique sequences; Zymo Research, USA) was added at four concentrations ($10^3$–$10^6$ copies/µL) for absolute quantification. The V3–V4 regions of the 16S rRNA gene were amplified using primers 338F (5′-ACTCCTACGGGAGGCAGCAG -3′) and 806R (5′-GACTACHVGGGTWTCTAAT-3′) with barcodes. Polymerase chain reaction (PCR) conditions: 95℃ for 3 min; 30 cycles of 95℃/30 s, 55℃/30 s, 72℃/45 s; final extension at 72℃ for 10 min. Reactions were performed in triplicate using a T100 Thermal Cycler (Bio-Rad, USA).

#### Library preparation and sequencing

Purified PCR products (QIAquick Gel Extraction Kit, Qiagen, Germany) were quantified with a Synergy HTX plate reader (Agilent, USA). Libraries were constructed using the NEXTFLEX Rapid DNA Seq Kit (Bioo Scientific, USA) and sequenced on the Illumina NextSeq2000 PE300 platform (Illumina, USA) at Shanghai Meiji Biomedical Technology Co., Ltd.

#### Bioinformatic analysis

Raw reads were processed using Fastp (v0.19.6) and FLASH (v1.2.11) for quality trimming, merging, and adapter removal. Operational taxonomic units (OTUs) were clustered at 97% similarity via UPARSE (v11), and taxonomic classification was performed using the RDP classifier (v2.13) against the SILVA database (v138). Absolute bacterial counts were calculated using spike-in standard curves. Alpha diversity (Shannon index) and Beta diversity were assessed using weighted UniFrac distance (for phylogenetic relatedness) and Bray-Curtis dissimilarity (for compositional differences). Principal Coordinate Analysis

(PCoA) was performed for visualization. Statistical significance of group separation was tested via permutational multivariate analysis of variance (PERMANOVA) with 999 permutations in QIIME2 (v2022.8).

## Non-targeted metabolomics analysis

### Metabolite extraction

Fecal metabolites were extracted from 50 mg samples using 400 µL acetonitrile:methanol (1:1, [vol/vol]; Sigma-Aldrich, USA). After vortexing (30 s), sonication (40 kHz, 30 min at 5°C), and centrifugation (13,000 × $g$, 15 min at 4°C), supernatants were dried under nitrogen gas and reconstituted in 120 µL acetonitrile:water (1:1).

### Liquid chromatography-tandem mass spectrometry analysis

Metabolites were separated on a Waters ACQUITY UPLC BEH C18 column (2.1 × 100 mm, 1.7 µm) using a UHPLC-Q Exactive HF-X system (Thermo Fisher Scientific, USA). Mobile phases: (A) 0.1% formic acid in water; (B) 0.1% formic acid in acetonitrile. Gradient: 0–2 min, 5% B; 2–15 min, 5%–95% B; 15–18 min, 95% B. MS parameters: ESI ± mode; resolution 70,000; scan range m/z 100–1,500.

### Data processing and pathway analysis

Raw data were processed using Progenesis QI (v3.0, Waters, USA) for peak alignment, normalization, and annotation against HMDB, Metlin, and KEGG databases. Multivariate analysis (principal component analysis [PCA] and orthogonal partial least squares discriminant analysis [OPLS-DA]) was performed in SIMCA-P (v14.1, Umetrics, Sweden). Differential metabolites (Variable Importance in Projection [VIP] >1, $P < 0.05$) were identified, and pathway enrichment was analyzed via Fisher's exact test.

## Statistical analysis

Wilcoxon rank-sum tests compared alpha diversity and bacterial abundances. Linear discriminant analysis effect size (LEfSe) identified differentially abundant taxa (linear discriminant analysis [LDA] score >2, $P < 0.05$). Correlations between microbiota and metabolites were assessed using Spearman's rank test (Python v3.9). Statistical significance was set at $P<0.05$.

Biomarker validation: Prevalence rates of significantly altered genera were calculated as the proportion of samples in each group where the genus was detected (detection threshold: >0 absolute counts). Receiver operating characteristic (ROC) curve analysis was performed using the pROC package in R (v4.3.1) to evaluate diagnostic potential. Area under the curve (AUC) values ≥0.7 were considered indicative of biomarker utility.

## RESULTS

The clinical characteristics of the two groups (age, gender, diet, medication history, etc.) showed no significant differences ($P > 0.05$; Table 1).

TABLE 1 Baseline characteristics of ARFC group and HC group

| Characteristic | ARFC group ($n = 16$) | HC group ($n = 15$) | $P$-value |
| --- | --- | --- | --- |
| Age (years) | 5.1 ± 1.3 | 4.9 ± 1.5 | 0.62 |
| Gender (male/female) | 10/6 | 13/8 | 0.89 |
| Dietary fiber intake (g/day) | 13.2 ± 2.7 | 12.9 ± 2.8 | 0.43 |
| Antibiotic use (past 3 months) | 0/16 | 0/21 | $-^a$ |

[a]–, not applicable (statistical test could not be performed as no antibiotic use was reported in either group).

## Analysis of GM diversity results of two groups of children

### There are differences in diversity between the two groups of children

Wilcoxon rank sum test for Chao index was applied to GM of the two groups, and the results showed that there was a difference in alpha diversity ($P < 0.05$; Fig. 1A). PCoA based on Bray-Curtis dissimilarity revealed a separation trend between groups ($P = 0.047$, PERMANOVA; Fig. 1B).

### There were significant differences in taxa between the two groups of children

The dysbacteriosis index of children in the two groups was significantly abnormal ($P < 0.001$; Fig. 2A). A Venn diagram is often used to show the number of shared and unique OTUs (operational taxa) or genes between two sample groups, including 1,547 OTUs shared by the two groups, 555 unique to the ARFC group, and 1,267 unique to the AR group (Fig. 2B). The absolute copy count of *Actinobacteria* in children in the ARFC group was significantly lower than that in the AR group ($P = 0.009$; Fig. 2C). The absolute abundance of multiple bacterial genera differed significantly between the two groups (Fig. 2D and Table 2). Notably, *Bifidobacterium* and *Blautia* were markedly reduced in the ARFC group ($P = 0.0125$ and $P = 0.0307$, respectively), whereas *Hungatella* was significantly elevated ($P = 0.0331$). Other genera such as *Ruminococcus_torques_group*, *Butyricicoccus*, and *Anaerostipes* also showed significant alterations (all $P < 0.05$), highlighting a broad disruption of gut microbiota composition in AR children with constipation. The underlying absolute quantitative data for all samples are provided in Table S2.

### Biomarker potential of Hungatella and Bifidobacterium

LEfSe multi-level taxonomic difference discriminant analysis (from phylum to genus level: phylum, class, order, family, and genus) carries out difference test at multiple levels, analyzes multi-level differential taxa, and uses LDA value to measure the impact of taxa on the difference effect, suggesting that these taxa may play a key role in the

1A

1B

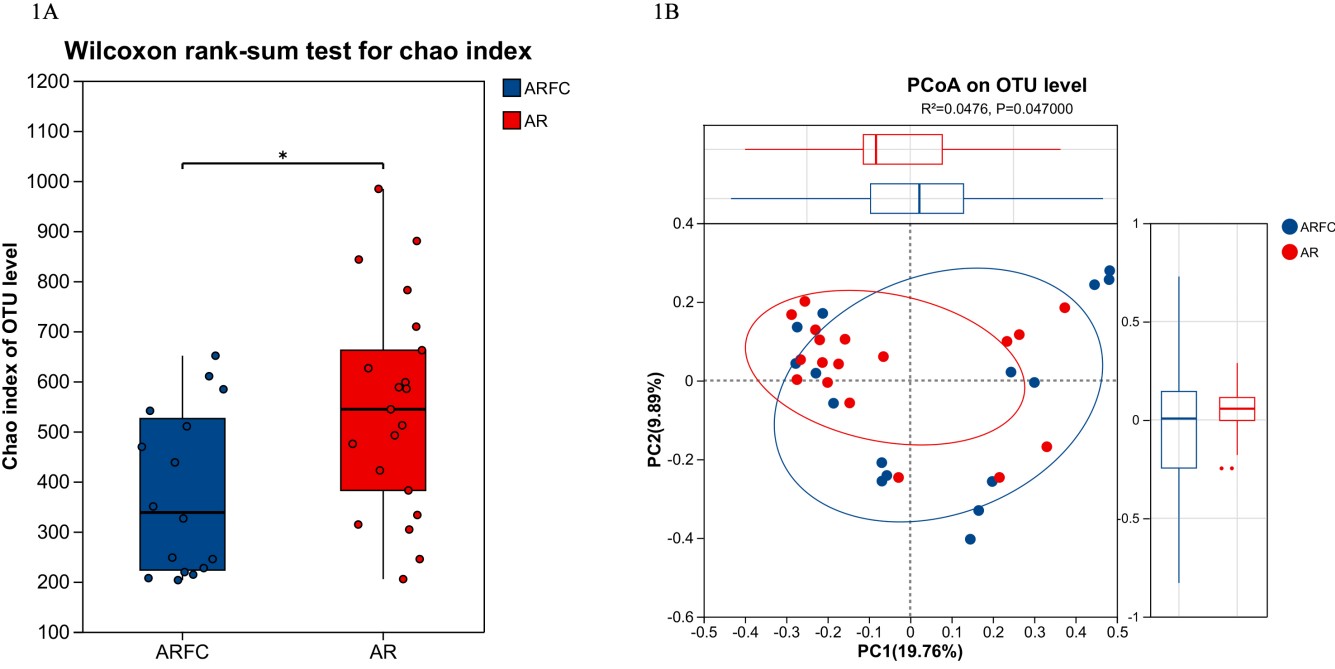

**FIG 1** Between-group microbial diversity comparisons. (A) Comparisons of the alpha diversity of GM in two groups, *P* values were calculated by using the Wilcoxon rank-sum test. (B) Using Principal coordinate analysis (PCoA) of β-diversity based on Bray-Curtis dissimilarity. Group separation was statistically significant ($P = 0.047$, PERMANOVA), presented by blue circle (ARFC) and red circle (AR).

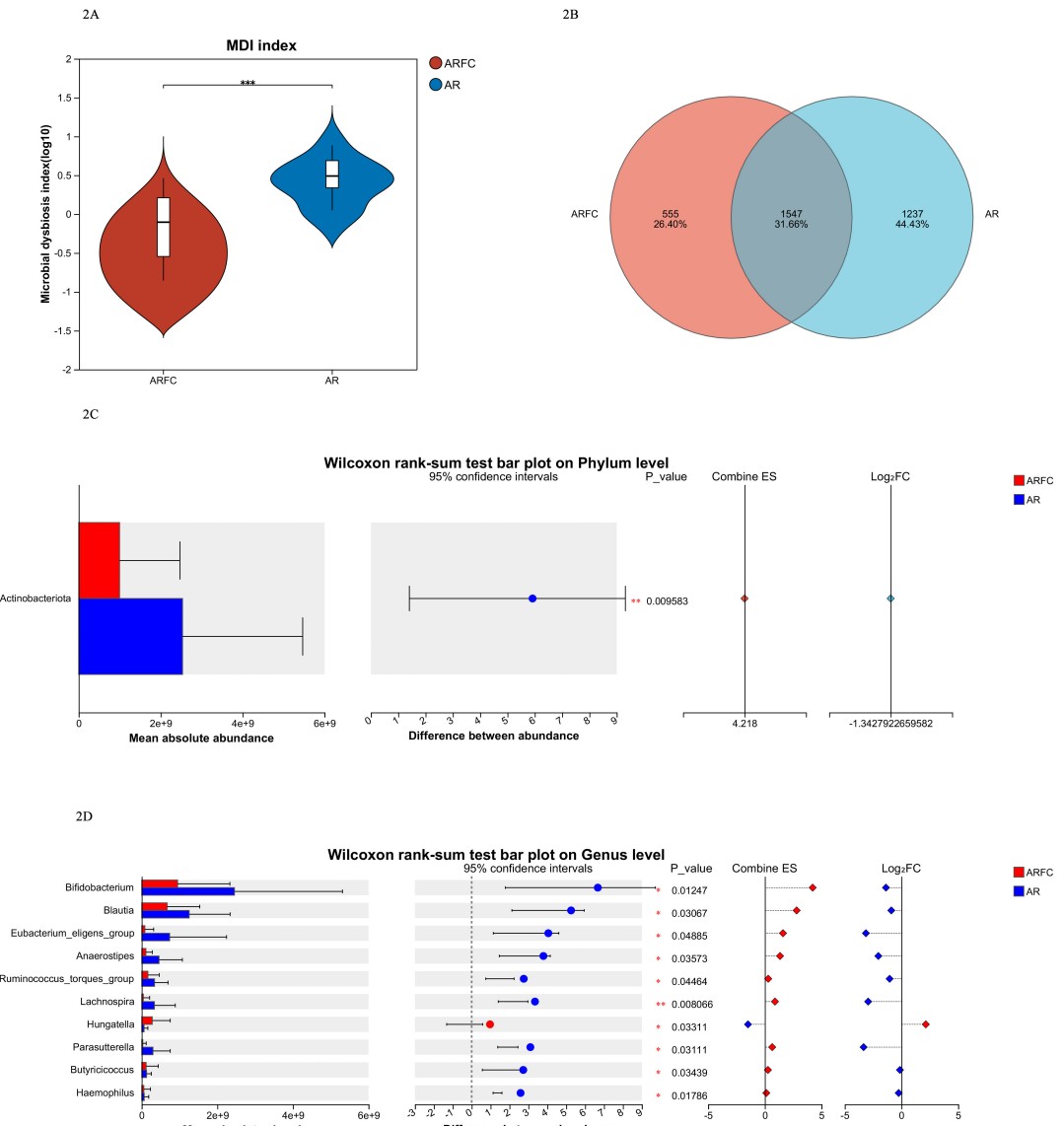

**FIG 2** Comparison of gut microbiota composition between two groups of children. (A) Comparison of gut microbiota dysbiosis index (MDI), red represents ARFC group, and blue represents AR group. (B) Comparison of the number of shared and unique OTUs in two sample groups. (C) Wilcoxon rank-sum test bar plot on the phylum level between two groups of children, the red bar chart represents ARFC group, and the blue bar chart represents the AR group. (D) Wilcoxon rank-sum test bar plot on genus level between two groups of children, the red bar chart represents the ARFC group, and the blue bar chart represents the AR group.

process of environmental change. The results showed that *Hungatella* (genus level) was significantly enriched in ARFC children, while *Bifidobacterium* (genus level) and other beneficial genera were enriched in AR children, and the above bacterial genera may be biomarkers of children in the two groups, respectively (Fig. 3A and B). LEfSe analysis identified *Hungatella* (genus level) as significantly enriched in ARFC children (LDA > 2, $P < 0.05$; Fig. 3C), suggesting its potential role as a microbial biomarker for constipation comorbidity in AR.

LEfSe identified *Hungatella* as significantly enriched in ARFC children (LDA > 2, $P < 0.05$; Fig. 3C). Prevalence analysis (Table 3 and Table S1) revealed that *Hungatella* was detectable in 94.1% (16/17) of ARFC samples compared to only 15.0% (3/20) in the AR group ($P < 0.001$). Conversely, *Bifidobacterium* was present in 95.0% of AR samples but only 29.4% of ARFC samples ($P < 0.001$). The detection rates of other beneficial genera

**TABLE 2** The top 15 dominant bacteria at the genus level in ARFC and AR groups (difference screening conditions: average relative abundance > 0.1%, $P < 0.05$, FDR < 0.05)

| The dominant bacteria (mean ± SD, %) | ARFC ($N = 16$) | AR ($N = 21$) | $P$-value |
|---|---|---|---|
| *Bifidobacterium* | $9.50 \times 10^8 \pm 1.38 \times 10^9$ | $2.45 \times 10^9 \pm 2.86 \times 10^9$ | 0.0125 |
| *Blautia* | $6.74 \times 10^8 \pm 8.62 \times 10^8$ | $1.26 \times 10^9 \pm 1.08 \times 10^9$ | 0.0307 |
| *Hungatella* | $2.80 \times 10^8 \pm 4.73 \times 10^8$ | $6.34 \times 10^7 \pm 9.91 \times 10^7$ | 0.0331 |
| *Ruminococcus_torques_group* | $1.65 \times 10^8 \pm 2.98 \times 10^8$ | $3.40 \times 10^8 \pm 3.56 \times 10^8$ | 0.0446 |
| *Butyricicoccus* | $1.18 \times 10^8 \pm 3.20 \times 10^8$ | $30 \times 10^8 \pm 1.27 \times 10^8$ | 0.0344 |
| *Anaerostipes* | $1.12 \times 10^8 \pm 1.68 \times 10^8$ | $4.59 \times 10^8 \pm 6.16 \times 10^8$ | 0.0357 |
| *Eubacterium_eligens_group* | $8.41 \times 10^7 \pm 2.33 \times 10^7$ | $7.40 \times 10^8 \pm 1.51 \times 10^9$ | 0.0489 |
| *Lachnospira* | $4.39 \times 10^7 \pm 1.66 \times 10^8$ | $3.37 \times 10^8 \pm 5.51 \times 10^8$ | 0.0081 |
| *Parasutterella* | $2.94 \times 10^7 \pm 8.97 \times 10^7$ | $2.98 \times 10^8 \pm 4.55 \times 10^8$ | 0.0311 |
| *Romboutsia* | $1.75 \times 10^7 \pm 4.12 \times 10^7$ | $4.02 \times 10^7 \pm 5.54 \times 10^7$ | 0.0197 |
| Lachnospiraceae_NC2004_group | $1.21 \times 10^7 \pm 3.45 \times 10^7$ | $4.27 \times 10^7 \pm 5.52 \times 10^7$ | 0.0163 |
| *Monoglobus* | $9.87 \times 10^6 \pm 2.14 \times 10^7$ | $6.38 \times 10^7 \pm 7.28 \times 10^7$ | 0.0128 |
| Ruminococcus_gauvreauii_group | $8.25 \times 10^6 \pm 2.80 \times 10^7$ | $3.53 \times 10^7 \pm 7.95 \times 10^7$ | 0.0248 |
| *Dorea* | $7.36 \times 10^6 \pm 2.85 \times 10^7$ | $7.46 \times 10^7 \pm 1.58 \times 10^8$ | 0.0018 |
| *Megasphaera* | $6.44 \times 10^6 \pm 2.58 \times 10^7$ | $1.46 \times 10^7 \pm 3.67 \times 10^7$ | 0.0040 |

such as *Blautia*, *Ruminococcus_torques_group*, and *Butyricicoccus* were also significantly lower in the ARFC group (all $P < 0.05$), reinforcing the widespread nature of the dysbiosis. These prevalence and abundance differences are underpinned by sample-level absolute quantitative data (Table S2). ROC curve analysis further supported their biomarker potential, with AUC values of 0.70 for *Hungatella* and 0.85 for *Bifidobacterium*, indicating

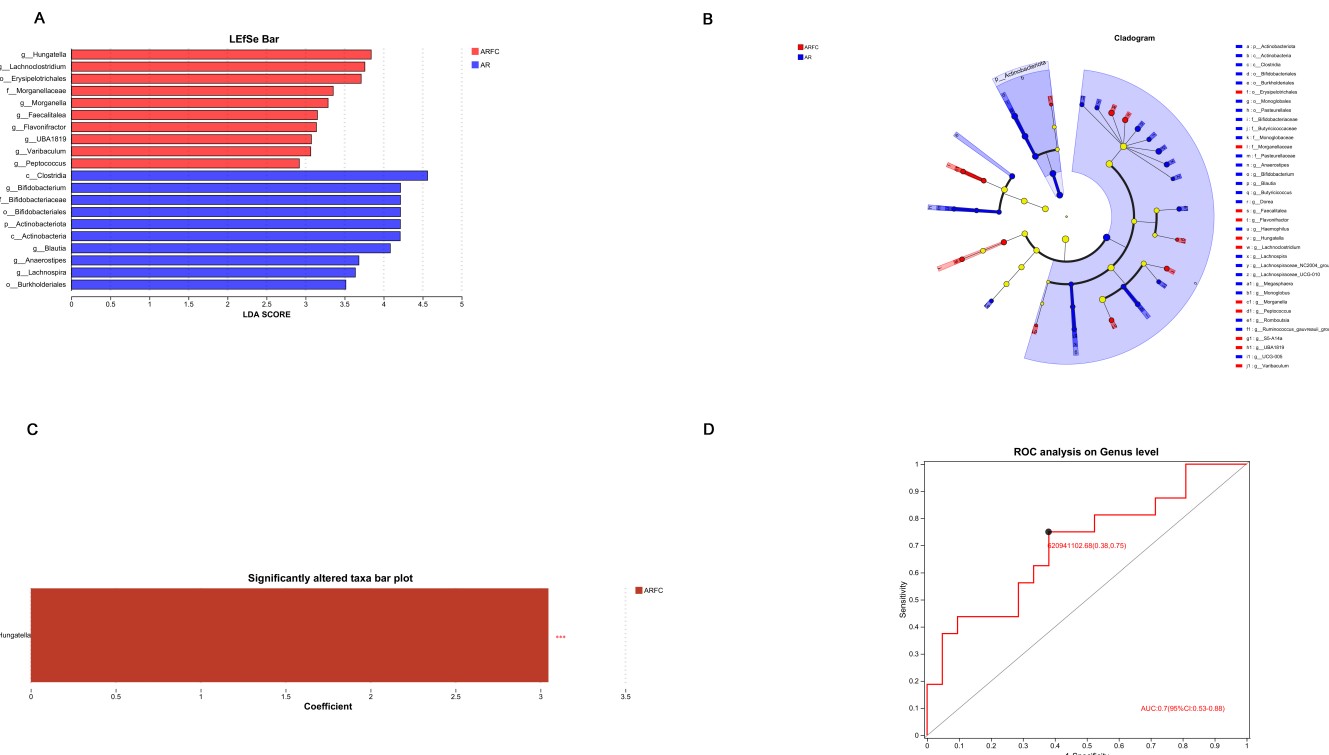

**FIG 3** Differential analysis of LEfSe multi-level taxa in the ARFC and AR groups of GM. (A) The default threshold for linear discriminant analysis (LDA) scoring is 2. The LDA discriminant bar chart is used to statistically analyze microbial communities with significant effects in multiple groups. The LDA score obtained through LDA analysis (linear regression analysis) indicates that the higher the LDA score, the greater the impact of taxa abundance on the differential effect. (B) Differentially enriched taxa (phylum to genus level) are color-coded by group. Light yellow nodes indicate non-significant taxa. (C) *Hungatella* enrichment identified by LEfSe (LDA > 2, $P < 0.05$). (D) ROC curves for *Hungatella* (AUC = 0.70, 95% CI: 0.53–0.88) and *Bifidobacterium* (AUC = 0.85).

**TABLE 3** Detection rates and AUC values of key differential bacterial genera between ARFC and AR groups

| Genus | Prevalence (ARFC) | Prevalence (AR) | P-value (prevalence) | AUC (ROC) |
|---|---|---|---|---|
| *Hungatella* | 94.1% (16/17) | 15.0% (3/20) | <0.001 | 0.70 |
| *Bifidobacterium* | 29.4% (5/17) | 95.0% (19/20) | <0.001 | 0.85 |

good discriminatory ability between groups. ROC analysis yielded an AUC of 0.70 (95% CI: 0.53–0.88) for *Hungatella* in distinguishing ARFC from AR children (Fig. 3D), indicating modest discriminatory capability and supporting its potential as an exploratory biomarker.

## Analysis of non-targeted metabolomics results in two groups of children

### Comparison of fecal metabolite composition and differences between the two groups of children

The metabolites in the 37 samples were extracted and then subjected to quality control, data pre-processing, and other steps. The number of common and unique metabolites in the two groups of children was counted through the Venn diagram, of which 3,093

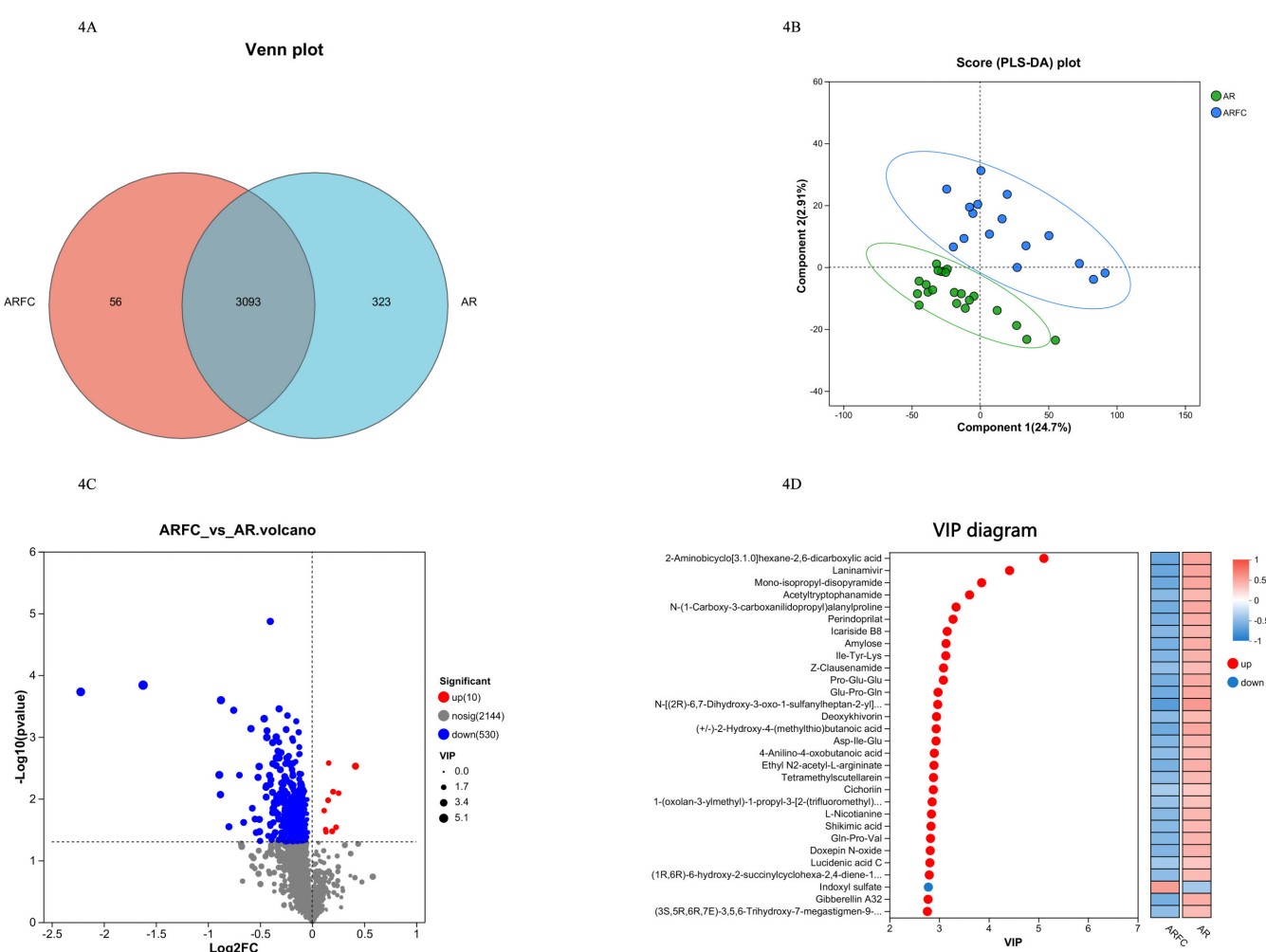

**FIG 4** Metabolite composition and differences between ARFC group and AR group. (A) Venn diagram showing the unique and shared metabolite in two groups. (B) Using principal component analysis to analyze metabolites in 37 fecal samples, presented by blue circle (ARFC) and green cube (AR). (C) Use tools to screen for differential metabolites between two groups and draw a volcano plot, with red metabolites upregulated in the ARFC group and blue metabolites downregulated. (D) Using OPLS-DA/PLS-DA as a supervised model, different changes in predicted pairings were tested through sevenfold cross-validation. VIP analysis was performed using the variable projection importance of the first principal component to obtain the results of VIP metabolites.

metabolites were shared, 56 were unique to the ARFC group, and 323 were unique to the HC group (Fig. 4A). According to the expression of metabolites among different samples, PCA was carried out on the samples to evaluate the similarity of samples within the group and the difference of samples between the groups (Fig. 4B), suggesting that the metabolites of children in the two groups were significantly different and dispersed. Then the VIP value of metabolites and fold change and *P*-value in univariate analysis were analyzed by OPLS-DA to screen the differential metabolites and draw a volcano diagram (Fig. 4C). Among them, 10 metabolites were up-regulated, and 530 were down-regulated in children in the ARFC group. OPLS-DA and partial least squares discriminant analysis (PLS-DA) were used as a supervised model to test and predict the different changes of pairing through sevenfold cross-validation. VIP analysis was carried out using the variable projection importance of the first principal component, and the important metabolites that help to classify were identified as markers to promote metabolism. Only indoxyl sulfate was up-regulated in ARFC children, and significantly down-regulated included 2-aminobicyclo(3.1.0)hexane-2,6-dicarboxylic acid, laninamivir, etc., while the opposite was true in the AR group (Fig. 4D).

### Comparison of metabolic pathways of metabolites between two groups of children

Compared with the AR group, the enrichment pathways of children in the ARFC group were mainly manifested in the abnormalities of amino acid metabolism and biosynthesis (including phenylalanine metabolism, D-amino acid metabolism, arginine and proline metabolism, valine, leucine, and isoleucine biosynthesis, phenylalanine, tyrosine, and tryptophan biosynthesis, etc.), as well as the digestion, absorption and synthesis of nutrients (biosynthesis of cofactors, protein digestion and absorption, vitamin digestion and absorption, mineral absorption, etc.), and others (such as biosynthesis of alkaloids derived from shikimate pathway, ABC transporters, etc.; Fig. 5).

### The bacteria with significant changes in the ARFC group were positively correlated with multiple amino acid metabolism

To further investigate the correlation between GM and metabolites in children with ARFC, we used Python 2.7.10 (v1.0) to analyze the correlation between significantly different bacterial genera and significantly different metabolites in the two groups of children. Procrustes analysis showed that the changes of GM and metabolites in the two groups of children were consistent and showed a separation trend (Fig. 6A). Mantel test network heatmap was used to analyze the significantly different bacterial genera and metabolites between the two groups of children. The results showed that the significantly different bacterial genera of the two groups of children were significantly positively correlated with amino acid metabolites (Fig. 6B).

## DISCUSSION

This study reveals distinct GM and metabolic profiles in AR children with constipation. The significant enrichment of *Hungatella* in the ARFC group aligns with prior reports linking this genus to intestinal motility inhibition via hydrogen sulfide production, a mechanism corroborated by its reduction post-fecal microbiota transplantation in constipation treatment (16, 17). Conversely, the depletion of *Bifidobacterium* and *Blautia* (Table S1)—critical producers of short-chain fatty acids (SCFAs) and regulators of amino acid metabolism (18, 19)—suggests a loss of microbial capacity to modulate intestinal homeostasis. This broad depletion, coupled with the near-ubiquitous presence of *Hungatella*, underscores a profound microbial imbalance that likely contributes to both constipation and immune dysregulation in this population. The detailed, sample-specific depletion of *Bifidobacterium* and enrichment of *Hungatella* (Table S2) provides quantitative evidence for this dysbiotic state. These findings are consistent with studies associating constipation with reduced SCFA levels and impaired gut barrier function (20, 21).

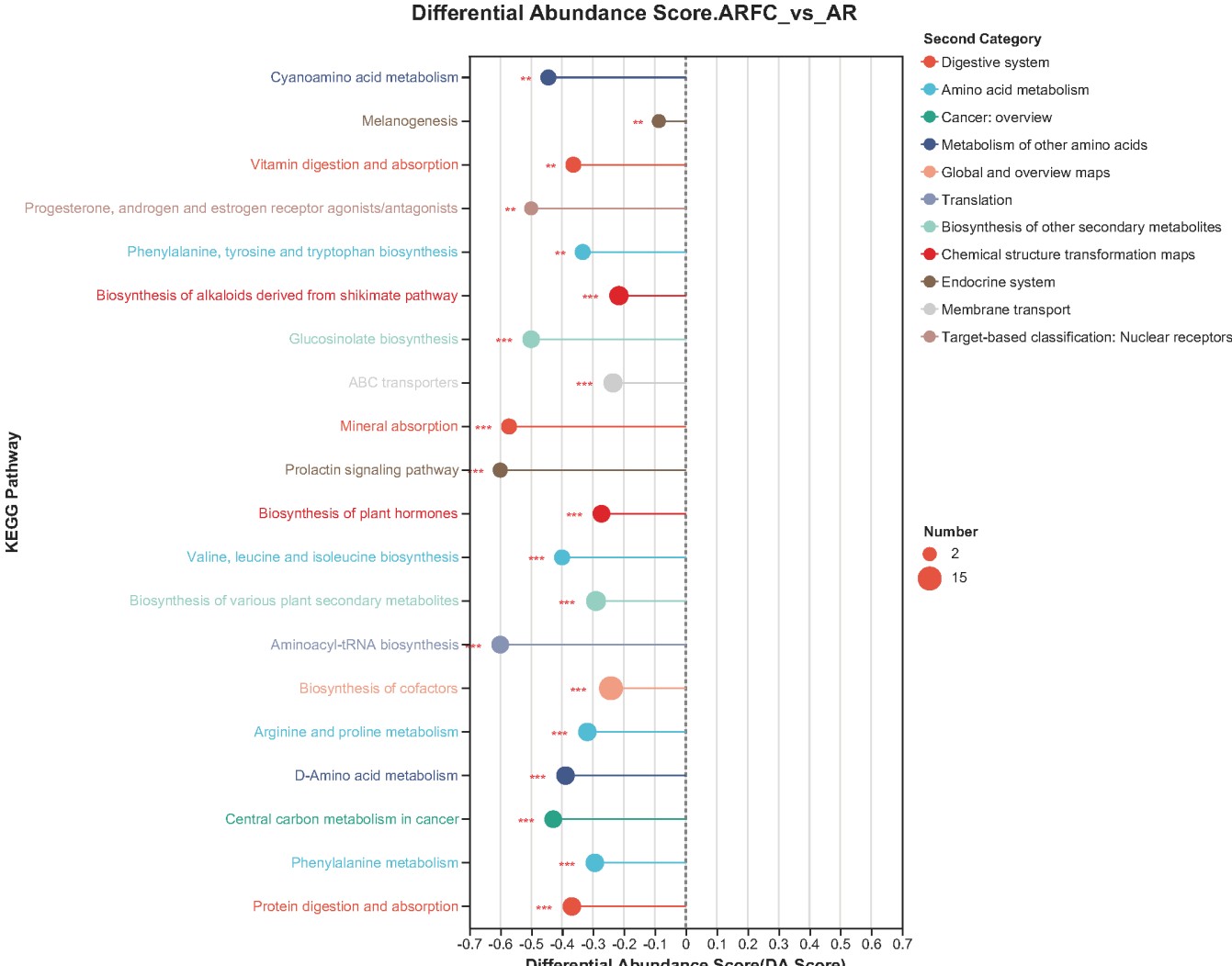

**FIG 5**  Comparison of metabolic product enrichment pathways between two groups, compare the metabolites generated from 37 samples to the KEGG pathway database to determine the KEGG functional pathways involved in all metabolites.

The metabolic shift toward indoxyl sulfate accumulation in ARFC children reflects aberrant tryptophan metabolism, likely driven by *Hungatella*-mediated indole production and subsequent hepatic sulfation (22, 23). Elevated indoxyl sulfate, a known activator of aryl hydrocarbon receptors (24), may exacerbate intestinal inflammation and immune dysregulation, potentially worsening both constipation and AR symptoms. Concurrent downregulation of aromatic and branched-chain amino acids underscores the role of GM in nutrient metabolism, as these amino acids serve as precursors for neurotransmitters and immune modulators (19). The positive correlation between dysbiotic genera and amino acid metabolites further supports a microbiota-metabolite axis in constipation pathogenesis.

While this study provides novel insights, limitations include a small sample size and a lack of mechanistic validation. Future work should expand cohorts, employ germ-free animal models to confirm causality, and explore interventions targeting *Hungatella* or probiotics like *Bifidobacterium* to restore metabolic balance.

## Conclusion and prospect

In conclusion, AR children with constipation exhibit a distinct GM profile characterized by *Hungatella* overgrowth and depletion of SCFA-producing genera, coupled with

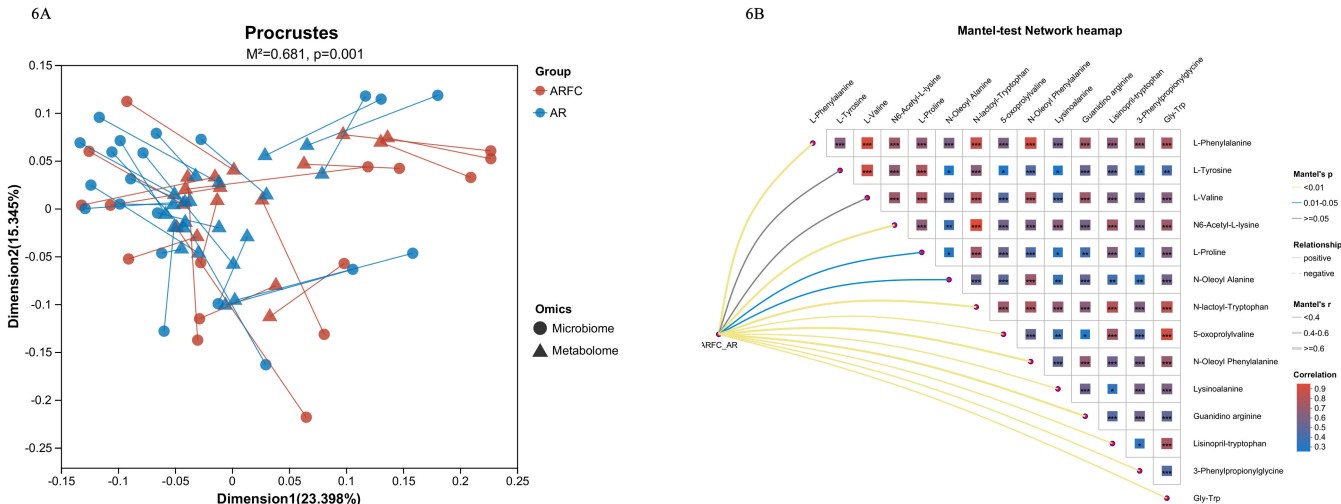

**FIG 6** The bacterial genera with significant changes in ARFC group children are positively correlated with the metabolism of multiple amino acids. (A) Procrustes analysis showed that the changes of GM and metabolites in the two groups were consistent, and there was a significant difference between the two groups (*P* = 0.001). (B) Mantel test network heatmap suggested that there was a significant positive correlation between GM and amino acid metabolites in ARFC children (mainly aromatic amino acids and branched-chain amino acids).

dysregulated amino acid metabolism and indoxyl sulfate accumulation. These findings position *Hungatella* and indoxyl sulfate as potential biomarkers for constipation in AR patients. The interplay between GM dysbiosis and metabolic disruption highlights the need for microbiota-targeted therapies, such as probiotics or dietary modifications, to ameliorate gastrointestinal and allergic symptoms. Future studies should validate these mechanisms in larger cohorts and preclinical models to advance personalized treatment strategies.

## ACKNOWLEDGMENTS

We thank all the participants for their support. We thank the doctors and nurses of Longgang District Maternity and Child Healthcare Hospital (Shenzhen, China) for assisting the research team in the clinical examination and fecal sample collection. We also thank the authors who made their data publicly available.

The research was generously supported by the Research Initiation Fund of Longgang District Maternity and Child Healthcare Hospital in Shenzhen City (Grant No. Y2024011), the Longgang District Science and Technology Innovation Bureau (Grant Nos. LGWJ2023-038 and LGWJ20230-072), and the Key Medical Disciplines Program in Longgang District.

Congfu Huang, Shen LV, and Weikeng Yang managed the project. Conceptualization: Congfu Huang and Shen LV. Data curation: Weikeng Yang and Chunyun Sun. Formal analysis: Weikeng Yang and Chunyun Sun. Funding acquisition: Congfu Huang. Investigation: Wei Kong. Methodology: Congfu Huang and Wei Kong. Project administration: Congfu Huang. Resources: Congfu Huang and Zhi Wan. Software: Wei Kong and Zhi Wan. Supervision: Zhi Wan. Validation: Weikeng Yang. Visualization: Ying Huang. Writing—original draft: Congfu Huang. Writing—review and editing: Shen LV and Ying Huang. All authors reviewed the manuscript.

This manuscript has not been published or presented elsewhere, either in full or in part, and is not under consideration for publication by any other journal.

All authors have reviewed and approved the content of this manuscript and consent to its submission to your respected journal.

## AUTHOR AFFILIATIONS

[1]Department of Pediatrics, Longgang Maternity and Child Institute of Shantou University Medical College (Longgang District Maternity and Child Healthcare Hospital of Shenzhen City), Shenzhen, China

[2]The Second Affiliated Hospital, CUHK-Shenzhen/Longgang District People's Hospital of Shenzhen, Shenzhen, China

[3]Longgang District Center for Disease Control and Prevention, Shenzhen, China

[4]The Second Clinical College of Southern Medical University, Guangzhou, China

## AUTHOR ORCIDs

Congfu Huang  http://orcid.org/0000-0002-3034-4459

## AUTHOR CONTRIBUTIONS

Shen Lv, Data curation, Methodology, Project administration, Writing – original draft, Writing – review and editing | Weikeng Yang, Formal analysis, Investigation, Methodology, Resources, Writing – review and editing | Chunyun Sun, Conceptualization, Data curation, Methodology, Software, Supervision | Ying Huang, Formal analysis, Software, Supervision, Validation, Visualization, Writing – review and editing | Wei Kong, Data curation, Investigation, Methodology, Supervision, Validation, Visualization | Zhi Wan, Formal analysis, Resources, Supervision, Validation | Congfu Huang, Conceptualization, Data curation, Funding acquisition, Methodology, Project administration, Resources, Writing – original draft, Writing – review and editing

## DATA AVAILABILITY

The raw GM data have been uploaded to the NCBI SRA database (PRJNA1143185), which will be accessible with the following link after the indicated release date.

## ETHICS APPROVAL

Ethical approval was obtained from the hospital's Institutional Review Board (No. KYXMLL-01-CZGC-14-2-1), and written informed consent was secured from all participants' guardians.

## ADDITIONAL FILES

The following material is available online.

### Supplemental Material

**Table S1 (Spectrum00748-25-S0001.xlsx).** Absolute quantitative abundance (copy number) of *Bifidobacterium* and *Hungatella* in fecal samples from children in the ARFC and AR groups.
**Table S2 (Spectrum00748-25-S0002.csv).** Comparative analysis of gut microbiota genera between AR and ARFC groups.

### Open Peer Review

**PEER REVIEW HISTORY (review-history.pdf).** An accounting of the reviewer comments and feedback.

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
