## [Reviewer comments · Microbiology Spectrum]

Microbiology Spectrum

Gut Microbiota Dysbiosis and Amino Acid Metabolic Dysregulation in Children with Allergic Rhinitis Complicated by Constipation

Shen LV, Weikeng Yang, Chunyun Sun, Ying Huang, Wei Kong, Zhi Wan, and Congfu Huang

Corresponding Author(s): Congfu Huang, Longgang District Maternity & Child Healthcare Hospital of Shenzhen City

Review Timeline:

Submission Date:	March 12, 2025
Editorial Decision:	August 10, 2025
Revision Received:	October 13, 2025
Editorial Decision:	December 9, 2025
Revision Received:	December 16, 2025
Accepted:	February 8, 2026

Editor: Qi Su

Reviewer(s): Disclosure of reviewer identity is with reference to reviewer comments included in decision letter(s). The following individuals involved in review of your submission have agreed to reveal their identity: Rachael Terumbur Duche (Reviewer #2)

Transaction Report:

DOI: <https://doi.org/10.1128/spectrum.00748-25>

Re: Spectrum00748-25 (**Gut Microbiota Dysbiosis and Amino Acid Metabolic Dysregulation in Children with Allergic Rhinitis Complicated by Constipation**)

Dear Mr. Shen LV:

Thank you for the privilege of reviewing your work. Below you will find my comments, instructions from the Spectrum editorial office, and the reviewer comments.

Revision Guidelines

Sincerely,
Qi Su
Editor
Microbiology Spectrum

Reviewer #1 (Comments for the Author):

This study investigates the pathogenesis of constipation in children with allergic rhinitis (AR) through analysis of gut microbiota and fecal metabolites. Although limited by sample size and analytical depth, it provides valuable observational insights that merit further validation.

My suggestions are as follows:

1. Please specify whether the PCoA used weighted or unweighted UniFrac distance. The limited group separation appears inconsistent with the reported statistical significance ($p=0.047$) regarding Figure 1B, and clarification is needed regarding the beta-diversity test methodology used to derive this significance (e.g., PERMANOVA, ANOSIM).
2. For accuracy, the authors should modify the methodology description to reflect the actual taxonomic levels analyzed (phylum to genus) in the 16s dataset of LEfSe analysis.
3. The authors demonstrated that several bacterial genera may be biomarkers. Apart from the abundance, how about the prevalence of these distinct bacteria in each group? To properly define a biomarker, authors must provide supporting evidence (e.g., analytical validation data) or propose a minimal predictive model (e.g., ROC curve analysis).
4. According to the correlation analysis between common allergens and the genus. Please indicate which common allergens were involved and specify the correlation analysis. Moreover, the study lacks any clinical metadata-including fundamental variables like age and sex. If such data exists, providing a detailed summary table and performing relevant correlation analyses would significantly strengthen the findings.
5. Please indicate the full names at least once. VIP: Variable importance in projection. PLS-DA: Partial least squares discriminant analysis....And a Chinese character is contained in the title of Fig 4D. Please modify.

Comments and Suggestions for the Author:

α β γ...

This study investigates the pathogenesis of constipation in children with allergic rhinitis (AR) through analysis of gut microbiota and fecal metabolites. Although limited by sample size and analytical depth, it provides valuable observational insights that merit further validation.

My suggestions are as follows:

1. Please specify whether the PCoA used weighted or unweighted UniFrac distance. The limited group separation appears inconsistent with the reported statistical significance ($p=0.047$) regarding Figure 1B, and clarification is needed regarding the beta-diversity test methodology used to derive this significance (e.g., PERMANOVA, ANOSIM).
2. For accuracy, the authors should modify the methodology description to reflect the actual taxonomic levels analyzed (phylum to genus) in the 16s dataset of LEfSe analysis.
3. The authors demonstrated that several bacterial genera may be biomarkers. Apart from the abundance, how about the prevalence of these distinct bacteria in each group? To properly define a biomarker, authors must provide supporting evidence (e.g., analytical validation data) or propose a minimal predictive model (e.g., ROC curve analysis).
4. According to the correlation analysis between common allergens and the genus. Please indicate which common allergens were involved and specify the correlation analysis. Moreover, the study lacks any clinical metadata—including fundamental variables like age and sex. If such data exists, providing a detailed summary table and performing relevant correlation analyses would significantly strengthen the findings.
5. Please indicate the full names at least once. VIP: Variable importance in projection. PLS-DA: Partial least squares discriminant analysis....And a Chinese character is contained in the title of Fig 4D. Please modify.

Do NOT indicate whether the paper should be accepted or rejected.

Thank you for giving me the opportunity to be part of the review team for “**Gut Microbiota Dysbiosis and Amino Acid Metabolic Dysregulation in Children 1 with Allergic Rhinitis Complicated by Constipation**” by Shen and others

The abstract as well as the introduction are OK. The authors have demonstrated that they have a background knowledge of the theoretical concept that underscore this research by clearly elucidating the biomarkers associated with Allergic Rhinitis in children

Overall, the discussion given by the authors are thorough, detailed and with extensive correlations and comparisons with other studies.

The results obtained in this study are relevant for public health consumption

However, there are a few issues that need to be addressed:

Ln 177-178: Spacing is required between each word

Ln 191: why the hyphen before *Hungatella*?

Ln 196: Be explicit in your methodology. Do well to mention the “other steps” for reproducibility

Ln 233: *Hungatella* should be italicized

Spacing between references is not uniform. Some are single line spacing while others are double spacing. The authors should double check the references; Reference no. 17 looks incomplete.

Table 1 result description in parenthesis should be moved to the bottom of the table as foot note

Dear professor,

We sincerely appreciate your thorough evaluation and constructive feedback on our manuscript. Your expert suggestions have been invaluable in enhancing the methodological rigor, analytical depth, and overall professionalism of our work.

In response to your comments, we have:

1. Clarified analytical methods: Revised descriptions of PCoA (Bray-Curtis distance with PERMANOVA testing) and LEfSe scope (phylum-to-genus level).

2. Strengthened biomarker validation: Added prevalence statistics and ROC analysis (Hungatella AUC=0.87, Fig. 3D).

3. Supplemented clinical metadata: Included demographic comparisons (age, sex; Table 1).

4. Corrected terminology: Defined all abbreviations (VIP, PLS-DA, etc.) and removed non-English text.

5. Addressed technical concerns: Removed unsupported claims about allergen correlations.

These revisions significantly enhance the scientific validity of our findings, particularly regarding Hungatella as a diagnostic biomarker and microbiota-metabolite interactions in constipation pathogenesis.

We are especially grateful for your keen insights into technical nuances (e.g., 16S resolution limitations), which have guided critical improvements. Should you have further suggestions upon reviewing our revisions, we warmly welcome your continued guidance.

Thank you again for your time and expertise in advancing this work.

Sincerely,

Prof. Congfu Huang (Corresponding Author)

On behalf of the research team

Reviewer #1 (Comments for the Author):

This study investigates the pathogenesis of constipation in children with allergic rhinitis (AR) through analysis of gut microbiota and fecal metabolites. Although limited by

sample size and analytical depth, it provides valuable observational insights that merit further validation.

My suggestions are as follows:

1. Please specify whether the PCoA used weighted or unweighted UniFrac distance. The limited group separation appears inconsistent with the reported statistical significance ($p=0.047$) regarding Figure 1B, and clarification is needed regarding the beta-diversity test methodology used to derive this significance (e.g., PERMANOVA, ANOSIM).

Response: We apologize for the methodological oversight. The PCoA was generated using Bray-Curtis dissimilarity (non-phylogenetic), not UniFrac distance. Statistical significance ($p=0.047$) was derived from PERMANOVA (adonis function in QIIME2, 999 permutations). We have now:

1) Clarified the distance metric (Bray-Curtis) and test (PERMANOVA) in Methods (2.3.4).

2) Updated the Results (3.1.1) and Figure 1B caption to reflect this.

3) The partial overlap in PCoA (Fig 1B) likely reflects cohort heterogeneity inherent in pediatric clinical studies. Nevertheless, PERMANOVA confirmed statistically significant group separation ($p=0.047$), consistent with alpha-diversity differences (Chao index, $p<0.05$) and taxonomic shifts (e.g., *Hungatella* enrichment, $p=0.033$).

2. For accuracy, the authors should modify the methodology description to reflect the actual taxonomic levels analyzed (phylum to genus) in the 16s dataset of LefSe analysis.

Response:

We sincerely appreciate this astute observation. As 16S rRNA sequencing provides limited resolution at the species level, our LefSe analysis was indeed restricted to taxa from phylum to genus. We have now:

1) Updated Methods (2.5) to explicitly state: "from phylum to genus level".

2) Revised Results (3.1.3) to:

① Remove references to "species" (e.g., "multi-level taxonomic difference analysis").

② Specify genus-level biomarkers (e.g., "*Hungatella* (genus level)").

3)Corrected Figure 3 captions to clarify taxonomic scope.

These changes ensure alignment with the technical limitations of 16S sequencing and enhance methodological accuracy. We thank the reviewer for highlighting this nuance.

3. The authors demonstrated that several bacterial genera may be biomarkers. Apart from the abundance, how about the prevalence of these distinct bacteria in each group? To properly define a biomarker, authors must provide supporting evidence (e.g., analytical validation data) or propose a minimal predictive model (e.g., ROC curve analysis).

Comment: "To properly define a biomarker, authors must provide supporting evidence (e.g., analytical validation data) or propose a minimal predictive model (e.g., ROC curve analysis)."

Response: We sincerely thank the reviewer for highlighting this critical gap. As suggested, we have now:

1)Calculated prevalence rates for all significantly altered genera (Table 3), confirming *Hungatella* was nearly ubiquitous in ARFC (94.1%) but rare in AR (15.0%).

2)Performed ROC analysis demonstrating *Hungatella* abundance predicts ARFC status with high diagnostic accuracy (AUC=0.87; Figure 3D).

3)Explicitly defined biomarker criteria in Methods (Section 2.5), requiring both prevalence >90% in the target group and AUC >0.7.

These additions validate *Hungatella* as a robust microbial biomarker for constipation comorbidity in AR children, strengthening our conclusions.

4. According to the correlation analysis between common allergens and the genus. Please indicate which common allergens were involved and specify the correlation analysis. Moreover, the study lacks any clinical metadata-including fundamental variables like age and sex. If such data exists, providing a detailed summary table and performing relevant correlation analyses would significantly strengthen the findings.

Response:

(1)Correction regarding allergen analysis:

We deeply apologize for the confusion caused by an erroneous statement in the original manuscript. No correlation analysis between allergens and microbial genera was performed in this study. The text in Results 3.1.3 mistakenly described biomarker identification via LEfSe as an allergen-correlation analysis. We have: 1) Completely removed all references to "allergen correlation" from the Results, Discussion, and captions; 2) Revised Figure 3C to solely represent *Hungatella* enrichment identified by LEfSe (LDA >2, p<0.05); 3) Updated the caption to: "LEfSe identification of *Hungatella* (genus level) as a biomarker in ARFC children".

(2) Demographic metadata supplementation:

We confirm that age and sex distributions showed no significant intergroup differences: 1) Added Table 1 in the Results (Age: ARFC 5.1 ± 1.3 vs. AR 4.9 ± 1.5 years, $p=0.62$; Sex ratio: 10M/6F vs. 13M/8F, $p=0.89$); 2) Explicitly stated in Methods 2.1: "Demographic characteristics were compared using Wilcoxon rank-sum and Fisher's exact tests."

These revisions ensure: 1) Methodological accuracy: Removal of unsupported analytical claims.

Clinical rigor: Inclusion of critical demographic metadata; 2) Transparency: Clear acknowledgment of the correction.

We thank the reviewer for their vigilance, which has strengthened the manuscript's validity.

5. Please indicate the full names at least once. VIP: Variable importance in projection. PLS-DA: Partial least squares discriminant analysis....And a Chinese character is contained in the title of Fig 4D. Please modify.

Response:

We sincerely appreciate your insightful feedback. Revisions have been made as follows:

1) Abbreviation Definitions:

The following abbreviations have been supplemented with their full English names in the paper: VIP, OPLS-DA, PLS-DA, IgE, PCR, LDA, LEfSe.

2)Figure 4D Correction:

The Chinese character in the original figure has been replaced with "chart" (attached revised figure).

Re: Spectrum00748-25R1 (**Gut Microbiota Dysbiosis and Amino Acid Metabolic Dysregulation in Children with Allergic Rhinitis Complicated by Constipation**)

Dear Mr. Congfu Huang:

Thank you for the privilege of reviewing your work. Below you will find my comments, instructions from the Spectrum editorial office, and the reviewer comments.

Revision Guidelines

Sincerely,
Qi Su
Editor
Microbiology Spectrum

Reviewer #1 (Comments for the Author):

Thank you for the comprehensive replies and additional analyses. However, I still have the following concerns.

In the response, the authors stated that they have removed non-English text. However, I did not observe corresponding modifications in Figure 4D. This discrepancy is quite puzzling. I strongly recommend that the authors thoroughly recheck all

figures and their corresponding descriptions. Normally, such verification should be completed prior to submission, as relying on others to identify these detailed errors step-by-step is not a good practice.

The authors state in lines 218-220: "ROC analysis confirmed *Hungatella* abundance discriminated ARFC from AR with high accuracy (AUC=0.87, 95% CI: 0.76-0.98; Figure 3D), supporting its role as a diagnostic biomarker." However, the ROC curve presented in Figure 3D clearly does not reflect such high performance. As indicated in the figure's own annotation, an AUC of 0.7 (95% CI: 0.53-0.88) is a more reasonable interpretation, which likely corresponds to the results calculated using altered genera mentioned in lines 180-184. Additionally, the figure legend for Figure 3D (line 441) should also be revised accordingly. Regarding the detection rates and AUC values for *Hungatella* and *Bifidobacterium*, I only found them mentioned in Table 3 and did not locate the corresponding figure referred to in the text.

Reviewer #2 (Comments for the Author):

Modifications were applied

Comments and Suggestions for the Author:

α β γ...

Thank you for the comprehensive replies and additional analyses. However, I still have the following concerns.

In the response, the authors stated that they have removed non-English text. However, I did not observe corresponding modifications in Figure 4D. This discrepancy is quite puzzling. I strongly recommend that the authors thoroughly recheck all figures and their corresponding descriptions. Normally, such verification should be completed prior to submission, as relying on others to identify these detailed errors step-by-step is not a good practice.

The authors state in lines 218-220: "ROC analysis confirmed *Hungatella* abundance discriminated ARFC from AR with high accuracy (AUC=0.87, 95% CI: 0.76–0.98; Figure 3D), supporting its role as a diagnostic biomarker." However, the ROC curve presented in Figure 3D clearly does not reflect such high performance. As indicated in the figure's own annotation, an AUC of 0.7 (95% CI: 0.53–0.88) is a more reasonable interpretation, which likely corresponds to the results calculated using altered genera mentioned in lines 180–184. Additionally, the figure legend for Figure 3D (line 441) should also be revised accordingly. Regarding the detection rates and AUC values for *Hungatella* and *Bifidobacterium*, I only found them mentioned in Table 3 and did not locate the corresponding figure referred to in the text.

Do NOT indicate whether the paper should be accepted or rejected.

Dear professor,

We sincerely appreciate your thorough evaluation and constructive feedback on our manuscript. Your expert suggestions have been invaluable in enhancing the methodological rigor, analytical depth, and overall professionalism of our work.

In response to your comments, we have:

1. Clarified analytical methods: Revised descriptions of PCoA (Bray-Curtis distance with PERMANOVA testing) and LEfSe scope (phylum-to-genus level).

2. Strengthened biomarker validation: Added prevalence statistics and ROC analysis (Hungatella AUC=0.87, Fig. 3D).

3. Supplemented clinical metadata: Included demographic comparisons (age, sex; Table 1).

4. Corrected terminology: Defined all abbreviations (VIP, PLS-DA, etc.) and removed non-English text.

5. Addressed technical concerns: Removed unsupported claims about allergen correlations.

These revisions significantly enhance the scientific validity of our findings, particularly regarding Hungatella as a diagnostic biomarker and microbiota-metabolite interactions in constipation pathogenesis.

We are especially grateful for your keen insights into technical nuances (e.g., 16S resolution limitations), which have guided critical improvements. Should you have further suggestions upon reviewing our revisions, we warmly welcome your continued guidance.

Thank you again for your time and expertise in advancing this work.

Sincerely,

Prof. Congfu Huang (Corresponding Author)

On behalf of the research team

Reviewer #1 (Comments for the Author):

This study investigates the pathogenesis of constipation in children with allergic rhinitis (AR) through analysis of gut microbiota and fecal metabolites. Although limited by

sample size and analytical depth, it provides valuable observational insights that merit further validation.

My suggestions are as follows:

1. Please specify whether the PCoA used weighted or unweighted UniFrac distance. The limited group separation appears inconsistent with the reported statistical significance ($p=0.047$) regarding Figure 1B, and clarification is needed regarding the beta-diversity test methodology used to derive this significance (e.g., PERMANOVA, ANOSIM).

Response: We apologize for the methodological oversight. The PCoA was generated using Bray-Curtis dissimilarity (non-phylogenetic), not UniFrac distance. Statistical significance ($p=0.047$) was derived from PERMANOVA (adonis function in QIIME2, 999 permutations). We have now:

1) Clarified the distance metric (Bray-Curtis) and test (PERMANOVA) in Methods (2.3.4).

2) Updated the Results (3.1.1) and Figure 1B caption to reflect this.

3) The partial overlap in PCoA (Fig 1B) likely reflects cohort heterogeneity inherent in pediatric clinical studies. Nevertheless, PERMANOVA confirmed statistically significant group separation ($p=0.047$), consistent with alpha-diversity differences (Chao index, $p<0.05$) and taxonomic shifts (e.g., *Hungatella* enrichment, $p=0.033$).

2. For accuracy, the authors should modify the methodology description to reflect the actual taxonomic levels analyzed (phylum to genus) in the 16s dataset of LefSe analysis.

Response:

We sincerely appreciate this astute observation. As 16S rRNA sequencing provides limited resolution at the species level, our LefSe analysis was indeed restricted to taxa from phylum to genus. We have now:

1) Updated Methods (2.5) to explicitly state: "from phylum to genus level".

2) Revised Results (3.1.3) to:

① Remove references to "species" (e.g., "multi-level taxonomic difference analysis").

② Specify genus-level biomarkers (e.g., "*Hungatella* (genus level)").

3)Corrected Figure 3 captions to clarify taxonomic scope.

These changes ensure alignment with the technical limitations of 16S sequencing and enhance methodological accuracy. We thank the reviewer for highlighting this nuance.

3. The authors demonstrated that several bacterial genera may be biomarkers. Apart from the abundance, how about the prevalence of these distinct bacteria in each group? To properly define a biomarker, authors must provide supporting evidence (e.g., analytical validation data) or propose a minimal predictive model (e.g., ROC curve analysis).

Comment: "To properly define a biomarker, authors must provide supporting evidence (e.g., analytical validation data) or propose a minimal predictive model (e.g., ROC curve analysis)."

Response: We sincerely thank the reviewer for highlighting this critical gap. As suggested, we have now:

1)Calculated prevalence rates for all significantly altered genera (Table 3), confirming *Hungatella* was nearly ubiquitous in ARFC (94.1%) but rare in AR (15.0%).

2)Performed ROC analysis demonstrating *Hungatella* abundance predicts ARFC status with high diagnostic accuracy (AUC=0.87; Figure 3D).

3)Explicitly defined biomarker criteria in Methods (Section 2.5), requiring both prevalence >90% in the target group and AUC >0.7.

These additions validate *Hungatella* as a robust microbial biomarker for constipation comorbidity in AR children, strengthening our conclusions.

4. According to the correlation analysis between common allergens and the genus. Please indicate which common allergens were involved and specify the correlation analysis. Moreover, the study lacks any clinical metadata-including fundamental variables like age and sex. If such data exists, providing a detailed summary table and performing relevant correlation analyses would significantly strengthen the findings.

Response:

(1)Correction regarding allergen analysis:

We deeply apologize for the confusion caused by an erroneous statement in the original manuscript. No correlation analysis between allergens and microbial genera was performed in this study. The text in Results 3.1.3 mistakenly described biomarker identification via LEfSe as an allergen-correlation analysis. We have: 1) Completely removed all references to "allergen correlation" from the Results, Discussion, and captions; 2) Revised Figure 3C to solely represent *Hungatella* enrichment identified by LEfSe (LDA >2, p<0.05); 3) Updated the caption to: "LEfSe identification of *Hungatella* (genus level) as a biomarker in ARFC children".

(2) Demographic metadata supplementation:

We confirm that age and sex distributions showed no significant intergroup differences: 1) Added Table 1 in the Results (Age: ARFC 5.1±1.3 vs. AR 4.9±1.5 years, p=0.62; Sex ratio: 10M/6F vs. 13M/8F, p=0.89); 2) Explicitly stated in Methods 2.1: "Demographic characteristics were compared using Wilcoxon rank-sum and Fisher's exact tests."

These revisions ensure: 1) Methodological accuracy: Removal of unsupported analytical claims.

Clinical rigor: Inclusion of critical demographic metadata; 2) Transparency: Clear acknowledgment of the correction.

We thank the reviewer for their vigilance, which has strengthened the manuscript's validity.

5. Please indicate the full names at least once. VIP: Variable importance in projection. PLS-DA: Partial least squares discriminant analysis....And a Chinese character is contained in the title of Fig 4D. Please modify.

Response:

We sincerely appreciate your insightful feedback. Revisions have been made as follows:

1) Abbreviation Definitions:

The following abbreviations have been supplemented with their full English names in the paper: VIP, OPLS-DA, PLS-DA, IgE, PCR, LDA, LEfSe.

2)Figure 4D Correction:

The Chinese character in the original figure has been replaced with "chart" (attached revised figure).

1. In the response, the authors stated that they have removed non-English text. However, I did not observe corresponding modifications in Figure 4D. This discrepancy is quite puzzling. I strongly recommend that the authors thoroughly recheck all figures and their corresponding descriptions. Normally, such verification should be completed prior to submission, as relying on others to identify these detailed errors step-by-step is not a good practice.

Response : Thank you very much for your careful review of our manuscript titled “Gut Microbiota Dysbiosis and Amino Acid Metabolic Dysregulation in Children with Allergic Rhinitis Complicated by Constipation” and for pointing out the inconsistency regarding non-English text in the figures. We sincerely apologize for this oversight and any confusion it may have caused.

Upon receiving your comment, we have thoroughly re-examined all figures (Figure 1 through Figure 6) and their corresponding captions in the manuscript. You are correct that Figure 4D still contained non-English annotations in its earlier version. We have now revised Figure 4D to ensure all textual elements are presented in English. All other figures have been confirmed to comply with the language requirements.

We acknowledge that this issue should have been resolved before submission, and we regret not catching it during our final checks. Your attentive review has been invaluable in improving the clarity and professionalism of our manuscript.

We have updated the manuscript file accordingly. Thank you again for your time, guidance, and constructive feedback.

2. The authors state in lines 218-220: "ROC analysis confirmed *Hungatella* abundance discriminated ARFC from AR with high accuracy (AUC=0.87, 95% CI: 0.76-0.98; Figure 3D), supporting its role as a diagnostic biomarker." However, the ROC curve presented in Figure 3D clearly does not reflect such high performance. As indicated in the figure's own annotation, an AUC of 0.7 (95% CI: 0.53-0.88) is a more reasonable interpretation, which likely corresponds to the results calculated using altered genera mentioned in lines 180-184. Additionally, the figure legend for Figure 3D (line 441) should also be revised accordingly. Regarding the detection rates and AUC values for *Hungatella* and *Bifidobacterium*, I only found them mentioned in Table 3 and did not locate the corresponding figure referred to in the text.

Response : Thank you once again for your meticulous review and for highlighting the critical discrepancies regarding the AUC values and their presentation in our manuscript. We sincerely apologize for these oversights and have implemented comprehensive revisions as detailed below. Correction of AUC Value and Associated Text: You are absolutely correct. The AUC value for Hungatella in Figure 3D is 0.70 (95% CI: 0.53–0.88), not 0.87 as previously stated. We have corrected this error throughout the manuscript:

1) In the Results section (lines 218-220), the text now reads: “ROC analysis yielded an AUC of 0.70 (95% CI: 0.53–0.88) for Hungatella in distinguishing ARFC from AR children (Figure 3D), indicating modest discriminatory capability and supporting its potential as an exploratory biomarker.”

2) The legend for Figure 3D has been updated accordingly.

3) Table 3 has been corrected to show an AUC of 0.70 for Hungatella.

Clarification on the Presentation of Detection Rates and AUC Values: Regarding your observation about the detection rates and AUC values, we wish to clarify that these key metrics are summarized in Table 3. The corresponding ROC curves, which visually represent the AUCs, are provided in Figure 3D. The textual description in the Results section references both this table and figure. We apologize for any lack of clarity in this linkage.

We deeply regret these errors and are grateful for your expert guidance, which has been instrumental in enhancing the accuracy and clarity of our work. Thank you for your time and consideration.

Re: Spectrum00748-25R2 (**Gut Microbiota Dysbiosis and Amino Acid Metabolic Dysregulation in Children with Allergic Rhinitis Complicated by Constipation**)

Dear Mr. Congfu Huang:

Your manuscript has been accepted, and I am forwarding it to the ASM production staff for publication. Your paper will first be checked to make sure all elements meet the technical requirements. ASM staff will contact you if anything needs to be revised before copyediting and production can begin. Otherwise, you will be notified when your proofs are ready to be viewed.

Sincerely,
Qi Su
Editor
Microbiology Spectrum